# Recent Advancements in the Mechanisms Underlying Resistance to PD-1/PD-L1 Blockade Immunotherapy

**DOI:** 10.3390/cancers13040663

**Published:** 2021-02-07

**Authors:** Yu Yuan, Abdalla Adam, Chen Zhao, Honglei Chen

**Affiliations:** 1Department of Pathology and Hubei Province Key Laboratory of Allergy and Immune-Related Diseases, School of Basic Medical Sciences, Wuhan University, Wuhan 430071, China; kelly_yuanyu@whu.edu.cn (Y.Y.); adamabdalla341@gmail.com (A.A.); 2Department of Oncology, Renmin Hospital of Wuhan University, Wuhan 430060, China; chen_zhao@whu.edu.cn

**Keywords:** PD-L1, resistance, immune checkpoints, immunotherapy

## Abstract

**Simple Summary:**

Immune checkpoint blockade targeting PD-1/PD-L1 has a promising therapeutic efficacy in different tumors, but a significant percentage of patients cannot benefit from this therapy due to primary and acquired resistance during treatment. This review summarizes the recent findings of PD-L1 role in resistance to therapies through the PD-1/PD-L1 pathway and other correlating signaling pathways. A special focus will be given to the key mechanisms underlying resistance to the PD-1/PD-L1 blockade in cancer immunotherapy. Furthermore, we also discuss the promising combination of therapeutic strategies for patients resistant to the PD-1/PD-L1 blockade in order to enhance the efficacy of immune checkpoint inhibitors.

**Abstract:**

Release of immunoreactive negative regulatory factors such as immune checkpoint limits antitumor responses. PD-L1 as a significant immunosuppressive factor has been involved in resistance to therapies such as chemotherapy and target therapy in various cancers. Via interacting with PD-1, PD-L1 can regulate other factors or lead to immune evasion of cancer cells. Besides, immune checkpoint blockade targeting PD-1/PD-L1 has promising therapeutic efficacy in the different tumors, but a significant percentage of patients cannot benefit from this therapy due to primary and acquired resistance during treatment. In this review, we described the utility of PD-L1 expression levels for predicting poor prognosis in some tumors and present evidence for a role of PD-L1 in resistance to therapies through PD-1/PD-L1 pathway and other correlating signaling pathways. Afterwards, we elaborate the key mechanisms underlying resistance to PD-1/PD-L1 blockade in cancer immunotherapy. Furthermore, promising combination of therapeutic strategies for patients resistant to PD-1/PD-L1 blockade therapy or other therapies associated with PD-L1 expression was also summarized.

## 1. Introduction

T-cell activation and proliferation induced by antigens is regulated by expression of both co-stimulatory and co-inhibitory receptors and their ligands [1]. Inhibitory pathways in the immune system can prevent autoimmunity through maintaining self-tolerance and regulating immunity [2]. While in tumors inhibitory pathways known as “checkpoints” can evade immune surveillance. Programmed cell death -1(PD-1) interacting with its corresponding ligand PD-L1 leads to immune suppression via preventing the T-cell activation in the tumor [3]. PD-1 is expressed on activated CD8^+^ T-cells as well as B cells and natural killer cells, and inhibits T-cell receptor (TCR) signaling and CD28 co-stimulation under chronic antigen exposure. As ligands of PD-1, PD-L2 is primarily expressed on antigen-presenting cells (APC) while PD-L1 is expressed on various types of cells including tumor cells and immune cells. Evidence of PD-L1 expression increase and spontaneous immune resistance is proved in several types of human cancers [4]. Besides, predictive and prognostic value of PD-L1 immunohistochemical expression has been reported in certain cancers. Moreover, PD-L1 as an inhibitory factor is also involved in other signaling pathways underlying mechanisms in resistance to tyrosine kinase inhibitors (TKIs).

Immunotherapy identified as the most promising approach in cancer treatment compared with chemotherapy and targeted therapy, immune checkpoint inhibitors have reported higher rates of response, remission, and better overall survival rates in a variety of tumors [5]. Immunotherapy has received the US Food and Drug Administration (FDA) approval for 57 indications in 17 solid tumors in less than 10 years, while over 80% are PD-1/PD-L1-targeted antibodies. Beneficial function of the PD-1/PD-L1 axis blockade is confirmed in treating many different types of cancers such as non-small cell lung cancer (NSCLC), melanoma and bladder carcinoma [6,7]. So far, six immune checkpoint inhibitors targeting PD-1/PD-L1 have been approved by the FDA for the first and second line of patients with non-small cell lung cancer including monoclonal antibodies (mAb) pembrolizumab, nivolumab and cemiplimab targeting PD-1 and mAb atezolizumab, avelumab and durvalumab targeting PD-L1. However, limited efficacy has been reported in PD-1/PD-L1 blockade therapy which rarely exceeds 40% in most cancer types and a large number of patients show partial responsiveness [8,9]. Even if there is a consistent rate of initial responses, the majority of patients develop therapeutic resistance and disease progression [10,11]. Focusing on PD-L1, we described all these concepts in this review including its predictive and prognostic value, immune resistance induced by PD-L1 and key mechanisms underlying resistance to PD-1/PD-L1 blockade therapy.

## 2. The Expression of PD-L1 Levels Predicting Resistance and Poor Prognosis

PD-L1 expression is increased in many types of human cancers and is regarded as a predictive and prognostic marker in cancer tissues. Prior data have demonstrated that PD-L1 expression is upregulated in cisplatin-resistant lung cancer cells compared with parent cells [12,13,14]. Resistance to epigenetic therapy is associated with enhanced PD-L1 expression in myeloid malignancies [15]. For example, 7 myelodysplastic syndrome and 6 acute myeloid leukemia patients received treatments with either azacytidine (Aza) or combined Aza and the histone deacetylase inhibitor LBH-589 to investigate the PD-L1 expression levels. Non-responders showed a more than two-fold increase in PD-L1 expression after treatment commenced, and except for two patients, none of the responders demonstrated increased expression of PD-L1.

PD-L1 expression is correlated with poor prognosis in different cancers. In chemotherapy and radiotherapy-treated patients with head and neck squamous cell carcinoma (HNSCC), high PD-L1 mRNA (>125 FPKM) from The Cancer Genome Atlas database had significantly reduced the 5-year survival rate [16]. Other data regarded PD-L1 as a potential biomarker for radiation therapy failure of HNSCC [17]. Following radiotherapy, a panel of radiation-resistant human papilloma virus (HPV)-negative HNSCC cell lines exhibited increased expression of PD-L1, three cohorts of HPV-negative HNSCC tumors with high expression of PD-L1 had much higher failure rates compared to the PD-L1-low expression group. Similar results have been reported in metastatic melanoma patients (MMP) [18]. Forty six and thirty four BRAFi-treated MMP harboring mutant BRAFV600 received vemurafenib and dabrafenib respectively. Patients with PD-L1 expression and an absence of tumor-infiltrating immune cells (TIMC) are related to shorter progression-free-survival compared to those with TIMC and absence of PD-L1. This study also identified PD-L1 overexpression and loss of TIMC as independent prognostic factors for melanoma-specific survival.

Interestingly, an experiment involving 18 patients with epidermal growth factor receptor (EGFR)-mutant NSCLC investigated the change of PD-L1 expression following gefitinib. A proportion of 38.9% (7/18) of NSCLC patients had a significant increase in the median H-score (marked as group A) of PD-L1, while the rest (61.1%) did not vary (group B). Besides, MET positivity by immunohistochemistry in biopsies is significantly correlated with group A. The results described a marked increase in expression of PD-L1 in tumor cells of a subset of patients after gefitinib treatment. Though EGFR-mutated NSCLC is prone to express less PD-L1 than wild type [19]. Similar results in several studies indicated that PD-L1 expression as a biomarker predicts resistance and poor prognosis after gefitinib treatment, rebiopsy should be considered [20]. Nevertheless, combination therapy with Durvalumab targeting PD-L1 and gefitinib has been proved to be more toxic and does not demonstrate a significant augmentation in progression-free survival (PFS) [21]. As a crucial factor predicting resistance and poor prognosis, PD-L1 has absolutely specific mechanisms for leading to resistance.

## 3. PD-L1-Induced Resistance

PD-L1 as an inhibitor in the immune system that induces immune resistance through interacting with its ligand PD-1. Besides, it is also involved in other signaling pathways generating resistance to TKIs.

### 3.1. PD-L1-Mediated Immune Resistance

In certain cancers, efficacy of antitumor treatment has always been found to be limited, due to the activation of immune checkpoints such as PD-1 and PD-L1. Once recognizing the tumor antigen, T-cells produce an anti-tumor immune response, which eventually leads to PD-1 lymphocyte expression and interferon release. To evade this immune attack, PD-L1 expression is adaptively upregulated by cancer cells and other inflammatory cells in the tumor microenvironment (TME) [22]. IFN-γ is secreted by tumor-infiltrating lymphocytes (TILs) and induces PD-L1 expression in the TME, thus T-cell cytotoxic function is impaired through the interaction of PD-L1 and PD-1. A similar pattern has been observed in other cancers including gastric cancer [23]. Fractionated radiation therapy can lead to increased tumor cell expression of PD-L1 in response to CD8^+^ T-cell production of IFN-γ [24]. In HPV-HNSCC, which is highly infiltrated by lymphocytes, IFN-γ-induced PD-L1 on tumor cells and CD68^+^ tumor-associated macrophages (TAMs) and highly expressed PD-L1 by CTLs, are found located at the same site [1].

In prior studies, PD-L1 expression is also upregulated followed by drug treatment and mediates an immune resistance. For example, in glioblastoma a compensatory recruitment of tumor-infiltrating myeloid cells elicited by antitumor immune response induced by dendritic cell (DC) vaccination contributed to the majority of PD-L1 expression [25]. Placenta-specific protein 8 (PLAC8) as an oncogene promoting cancer growth and progression is abnormally upregulated in gallbladder carcinoma. Overexpression of PLAC8 conferred resistance to gemcitabine and liplatin (OXA), mainstays of chemotherapy by upregulating PD-L1 expression [26]. 5-Fluorouracil selectively depletes myeloid-derived suppressor cells (MDSCs) and OXA triggers an immunogenic form of tumor cell death. A combined chemotherapy Folfox, 5-Fluorouracil plus OXA, has routinely been regarded as a first line of treatment for advanced colorectal cancer. However, Folfox up-regulates high expression of PD-1 on activated CD8^+^ TILs, and induces CD8^+^ T-cells to secret IFN-γ which upregulates PD-L1 expression on tumor cells [27]. CD40 stimulation on APC directly activates CTLs without the help of CD4^+^ T-cells. Agonistic anti-CD40 antibodies induce antitumor responses and upregulation of PD-L1 on tumor-infiltrating monocytes and macrophages, which are extremely dependent on T-cells and IFN-γ [28]. When co-cultured with human PBMC, trastuzumab, the anti-human epidermal growth factor receptor-2 (HER2) antibody, is shown to upregulate PD-L1 in HER2-overexpressing breast cancer cells via mediating stimulation of IFN-γ secretion on immune cells [29]. Inhibitors of mTOR approved by the Food and Drug Administration to treat advanced metastatic renal cancers and enhance nuclear translocation of transcription factor EB, was bound to PD-L1 promoter and thereby led to increased PD-L1 expression [30].

### 3.2. Signaling Pathways and Factors Involved in PD-L1-Induced Resistance

Despite immune resistance, PD-L1 has generated resistance to TKIs in certain cancers. Possible mechanisms by which PD-L1 induced acquired resistance through upregulating Yes-associated protein1 (YAP1), [31] Bcl-2-associated athanogene-1 (BAG-1), [32] and DNA methyltransferase 1 (DNMT1), [33] and generated primary resistance by inducing epithelial-to-mesenchymal transition (EMT) have been reported [34] (Figure 1).

Activation of MEK/extracellular single-regulated kinase (ERK) signaling furthers phosphorylation and ubiquitination of the Bcl-2-interacting mediator of cell death (BIM), a BH-3-only protein, thereby preventing cells from apoptosis [35]. Resistance to TKI in NSCLC generally occurs through reactivating ERK signaling [36]. EGFR mutation activation induces expression of PD-L1 in NSCLC cells via ERK-signaling [37]. Once triggered by ERK signaling, phosphorylated C/EBPβ induced by PD-L1 can enhance binding to the BAG-1 promoter, thus promoting BAG-1 expression. The PD-L1/BAG-1 axis confers TKI resistance through persistent activated ERK signaling via the EGFR/ERK/PD-L1/BAG-1 feedback loop [32]. Thus combining treatment with TKIs and anti-PD-L1 therapy may provide a promising strategy for tumors with a high expression of PD-L1 and BAG-1, though this has not been researched yet.

YAP1 is another factor known to confer EGFR-TKI resistance in lung cancer cells [38]. Distinct experiments utilizing reactive oxygen species (ROS) scavengers and inducers demonstrated a concomitant change of expression of PD-L1 and hypoxia-inducible factor-1α (HIF-1α), YAP1 [31]. While prior reports described that PD-L1-induced HIF-1α is stimulated by the generation of ROS [39,40], hypoxia promotes formation of YAP1 and HIF-1α complex via regulating SIAH2 ubiquitin E3 ligase and increases YAP1 gene expression [41,42]. TKI resistance may be conferred by PD-L1/ROS/HIF-1α/YAP1 axis and a YAP1/EGFR/ERK/NF-κB loop [31]. Markedly high expression of YAP and PD-L1 are observed in EGFR-TKI-resistant cells in another study, and they demonstrate a positively related change in expression when given a knockdown of YAP [43]. Thereby, giving an anti-PD-L1 or anti-YAP1 may overcome the EGFR-TKI resistance.

The PD-L1/DNMT1 axis is also a critical mechanism leading to acquired resistance [33]. DNMT1, as a member of the DNA methyltransferase family, maintains the DNA methylation pattern [44]. Signal transducer and activator of transcription 3 (STAT3), a well-characterized transcription factor that binds to DNMT1 promoter and positively regulates transcription of DNMT1 [45], since phosphorylated STAT3 induces transcriptional activation via binding with specific DNA elements. PD-L1 regulates DNMT1 through the STAT3-signaling pathway and induces DNMT1-dependent DNA hypomethylation to promote development of cancers [46], thereby resulting in acquired resistance [33]. Currently, a combination therapy with oxaliplatin and decitabine inhibiting DNA demethylation was proved to have a synergistic effect in enhancing anti-PD-L1 therapeutic efficacy in colorectal cancer [47].

The transforming growth factor-beta (TGF-β)/Smad signaling pathway plays a role in PD-L1-induced primary resistance to EGFR-TKIs [34]. EMT can decrease efficacy of drug treatment in NSCLC [48,49]. PD-L1 upregulates phosphorylation of Smad3, which significantly participates in the transcriptional regulation mediated by TGF-β1 [50], and the TGF-β/Smad-signaling pathway has been reported to be crucial in EMT progression [51]. The mechanism of primary resistance to EGFR-TKIs in EGFR-mutant NSCLC may confer through the PD-L1/TGF-β/Smad/EMT axis [34]. In addition, in Kirsten rat sarcoma viral oncogene homolog (KRAS)-mutant NSCLC, KRAS G12 mutation is reported to promote PD-L1 expression via a TGF-β/EMT-signaling pathway [52]. Apparently, PD-L1 expression plays a key role in poor prognosis and resistance after treatment in several types of cancers, thereby adding an anti-PD-1 or anti-PD-L1 therapy may improve the efficacy and become a promising therapeutic strategy.

## 4. Key Mechanisms Underlying Resistance to PD-1/PD-L1 Blockade

PD-1/PD-L1 blockade therapy has been approved as a significantly helpful treatment in certain cancers, a problem of its limited efficacy has occurred and the targeting solution is urgently discussed and provided. Focusing on PD-L1, we described key mechanisms underlying resistance to PD-1/PD-L1. Surprisingly, abnormally upregulated PD-L1 expression and a lack of PD-L1 can both lead to inefficacy of PD-1/PD-L1 inhibitors (Figure 2).

### 4.1. Aberrant PD-L1 Expression

PD-L1 is generally regulated by tumor cells in two ways: the first is innate immune resistance in which constitutive oncogenic signaling is correlated with PD-L1 expression, the second is an adaptive immune resistance through which IFN-γ produced by TILs induces PD-L1 expression.

K-ras mutation as a common oncogenic driver in the lung adenocarcinoma (LUAD) and upregulates PD-L1 through p-ERK instead of p-AKT signaling [53]. Different subgroups of KRAS-mutant LUAD are dependent on STK11/LKB1 or TP53 mutations, and alterations of the former has been confirmed as a major factor that leads to primary resistance to PD-1 blockade [54]. Besides, EGFR-mutant or ALK-rearranged patients had a PD-L1 tumor proportion score of ≥50% and turned out not to respond to PD-1/PD-L1 inhibitors [55].

The transcription factor Yin Yang 1 (YY1); a major regulator reported participating in various pathways, is involved in cell growth, survival and metastasis. YY1 upregulates PD-L1 expression on tumor cells via signaling pathways, including p53, STAT3, NF-κB and PI3K/AKT/mTOR [56]. PD-L1v242 and PD-L1v229, two secreted PD-L1 C-terminal splicing variants, could capture the aPD-L1 antibody and function as a “decoy” to prevent antibodies from binding to PD-L1 [57].

Besides, a tumor-intrinsic signaling pathway involved with NLRP3 inflammasome in response to upregulated expression of PD-L1 was found to drive adaptive resistance to anti-PD-1 antibody immunotherapy [58]. NLRP3 inflammasome triggered by PD-L1 induces tumor Wnt5α expression via HSP70-TLR4 signaling, while non-canonical WNT ligands promote production of CXCR2 ligands through the activated YAP pathway [59,60]. CXCR2-relied migration and recruitment of a granulocytic subset of MDSCs (PMN-MDSCs) play a role in suppressing CD8^+^ T-cell infiltration and function, therefore leading to adaptive resistance [61,62].

Previous study showed that tumors can be divided into four categories according to positive/negative tumor PD-L1 expression and presence/absence of TILs. For instance, patients with PD-L1 positive and TILs indicate adaptive immune resistance and those with PD-L1 negative and without TILs show immune ignorance [63]. Among these four types, type I with PD-L1 positive and TILs is the most likely to respond to PD-1/PD-L1 blockade therapy, whilst other types may show unresponsiveness to this monotherapy [64].

### 4.2. Paucity of PD-L1 Expression

The interaction between PD-L1 and its receptor PD-1 leads to immune escape and inhibits T-cell function and blockade of PD-L1 and PD-1 enhances the antitumor immunity in several cancers. However, the expression of PD-L1 or PD-1 is a prerequisite for the therapeutic efficacy. Evidence of the relation of rare PD-L1 expression and poorer responses to PD-1 blockade has been proved in prostate cancer [65]. DNA hypomethylating agent upregulate PD-L1 gene expression [66]. Anti-PD-1 therapy curbs the expression of PD-L1 through either eliminating the tumor cells that overexpress PD-L1 and possess a hypomethylated PD-L1 promoter or switching off the PD-L1 expression through epigenetic modulation, therefore leading to resistance [67]. Loss-of-function mutations in JAK1/2 can lead to primary resistance to anti-PD-1 therapy due to the inability to respond to IFN-γ for a lack of PD-L1 expressions [68]. Despite the effect of aberrant PD-L1 expression, an abnormal process from antigen expression to T-cell activation can result in resistance to PD-1/PD-L1 inhibitors. Moreover, a recent study demonstrated that PD-L1 expression is enhanced via nicotinamide adenine dinucleotide (NAD^+^) metabolism, in which nicotinamide phosphoribosyltransferase (NAMPT) functions as the rate-limiting enzyme [69]. NAMPT increases PD-L1 expression induced by IFN-γ and leads to immune escape in tumors with the help of CD8^+^ T-cells. Thus NAD^+^ metabolism is a promising strategy for resistance to anti-PD-L1 therapy [69].

### 4.3. Aberrant Antigen Expression, Presentation and Recognition

Tumors with a higher tumor mutation burden (TMB) are likely to have more neoantigens, which can be recognized by the immune system as “non-self” in response to checkpoint inhibition. In Naiyer’s study, the result of the treatment of PD-1 targeting antibody pembrolizumab in NSCLC described that a higher burden of nonsynonymous tumors is correlated with a better response and PFS [70]. Besides, strong immunogenicity and extensive expression of immune checkpoint ligands make the microsatellite instability subtype more susceptible to immunotherapeutic methods, for example, with anti-PD-L1 and anti-cytotoxic T lymphocyte-associated antigen-4 (CTLA-4) antibodies [71]. Tumors with defective mismatch repair possess more DNA mutations and show an improved responsiveness to anti-PD-1 therapy [72]. In short, a low mutational burden, microsatellite stability and efficient DNA repair mechanisms are involved in innate resistance to immune-checkpoint blockade therapy. Moreover, evolution of neoantigen loss can produce an acquired resistance [73]. A study also demonstrates that deficiency of heterogeneity in HLA genes is observed in cancer development, a high level of HLA loss results in acquired resistance during immunotherapy [74].

Resistance to immune checkpoint blockades also involves impaired DC maturation, which is an essential process in T-cell activation, through it is displayed in various co-stimulatory factors expression including MHC class I/II, CD80, CD86 and CD40 [75]. IL37b decreases CD80 and CD86 expression through the ERK/S6K/NF-κB axis and suppresses DC maturation [76]. A transcription factor STAT3 that facilitates tumor growth and metastasis leads to the induction of other immunosuppressive factors that possess a suppressive function on DC maturation, including IL10, Tregs and TGF-β [77,78,79,80].

Despite inducing PD-L1, IFNs have been reported to (re-)activate T-cells to control the tumor development via advancing DC cross-priming [81,82,83]. It is well-known that CTLs recognize MHC class I-presented peptide antigens on the surface of tumor cells. Heterozygous mutations, deletions or deficiency in β-2-microglobulin (β2M); a crucial factor in MHC class I antigen presentation, generally reduces antigen recognition by antitumor CD8^+^ T-cells and mutation of β2M gene leads to resistance to anti-PD-1 therapy [84,85]. IFN-γ can induce tumor cells to express MHC class I molecules, significantly promoting CTL differentiation and enhancing apoptosis. Mutations or loss of IFN-γ pathway-related proteins on tumor cells (such as STATs, IFN-γ receptor chain JAK1 and JAK2) can cause escape from immune recognition and resistance to immune checkpoint inhibitions [68,86].

### 4.4. Aberrant Immunity of T-Cells

Despite normal antigen expression, presentation, recognition and successfully activated T-cells, resistance to the PD-1/PD-L1 blockade inhibitors may occur owing to the T-cell itself. The aberrant immunity of T-cells include insufficient T lymphocytes infiltration, dysfunction of T-cell and exhausted T-cells.

#### 4.4.1. Insufficient T Lymphocytes Infiltration

Despite the expression of PD-L1, a lack of T lymphocyte infiltration can cause unresponsiveness to anti-PD-L1 therapy. A crucial prerequisite for the therapeutic efficacy is the existing and tumor-infiltrated anti-tumor CTLs [87]. LIGHT, a member of the tumor necrosis factor superfamily, may activate lymphotoxin β-receptor signaling, resulting in the generation of chemokines that recruit a huge number of T-cells [88].

The PI3K-AKT-mTOR pathway, a crucial oncogenic signaling pathway, is involved in a multitude of cellular processes including cell survival, proliferation, and differentiation. PTEN, a lipid phosphatase, inhibits the PI3K signaling activity which activates the pathway. Loss of PTEN has been reported to reduce CD8^+^ T-cells infiltrating into tumors and lead to resistance to PD-1 blockade therapy. A selective PI3Kβ inhibitor treatment enhanced the efficacy of anti-PD-1 antibodies [89]. The MAPK pathway also plays a major role in cell proliferation, inhibits T-cell recruitment and functions by inducing VEGF and IL-8 [90]. An inhibited MAPK pathway promotes CD8^+^ T-cell activation and infiltration in melanoma [91,92]. Furthermore, studies showed that the combination of PD-1 blockade, BRAFi and MEKi enhances tumor immune infiltration and improves treatment outcomes [93].

A crucial oncogenic signaling pathway Wnt/β-catenin has been highly related to immune escape [94,95]. An activated Wnt/β-catenin pathway is correlated to loss of T-cell gene expression in metastatic melanoma [96]. Another study reported that the activation of the Wnt/β-catenin pathway in tumors brings about a non-inflammatory environment via numerous mechanisms. For instance, it acts on CD103^+^ DCs of the Batf3 lineage and induces the transcription inhibitor ATF3 (activating transcription factor 3) expression to decrease production of Chemokine (C-C motif) ligand 4 (CCL4), thereby reducing initiated and infiltrated CTLs. Moreover, the Treg survival rate is enhanced by β-catenin [97].

Recently immune tumors are divided into three phenotypes: immune-desert, excluded and inflamed. Among these, the first and second phenotypes, which are non-inflamed tumors, show a low density of CTLs in the tumors and poor prognosis in immune checkpoint blockade therapy [98].

#### 4.4.2. Dysfunction of T-Cells

Accumulation of extracellular adenosine is exploited by tumors to escape immunosurveillance through the activation of purinergic receptors [99]. CD38 expression expressed on Tregs and MDSCs is infiltrated in the tumor microenvironment and stimulated adenosine production via the CD38–CD203a-CD73 axis, and therefore inhibits CTL function [100,101].

#### 4.4.3. Exhausted T-Cells

In vitro studies have reported that the PD-1 signal intensity determines the severity of T-cell exhaustion, which in turn affects the efficacy of anti-PD-1 treatment. In Nigow’s animal model, high expression of PD-1 and extremely unresponsive T-cells showed relevance with resistance of anti-PD-1 therapy [102]. PD-1 treatment helps patients with low or moderate PD-1 expression to re-invigorate exhausted CD8^+^ T cells and exert their immune effects. However, the cellular, transcriptional, and epigenetic changes following the PD-1 pathway blockade suggested limited storage potential after TEX re-invigoration, which means re-exhaustion following PD-L1 blockade [103].

## 5. Tumor-Suppressing Microenvironment

Apart from abnormal T-cells and PD-L1 expression, there are some other types of cells and cytokines that benefit tumor development inside the tumor microenvironment, they form the tumor-suppressing microenvironment to play a key role in resistance to the PD-1/PD-L1 blockade.

### 5.1. Tregs

Tregs are involved in maintaining self-tolerance, and inhibit autoimmunity through secreting cytokines, including TGF-β1. The ratio of CD8^+^ Teff cells/Tregs is strongly associated with the prognosis of immunotherapy [104,105]. The administration of low-dose TLR-7 agonist resiquimod could transform Treg accumulation-caused resistance to the PD-L1 blockade [106]. Combination of radiation therapy and dual immune checkpoint blockade restores antitumor immunity of consumed Tregs [107]. Currently, anti-CD25 therapy is believed to take effect through Treg depletion when combined with PD-1 blockade therapy [108].

### 5.2. MDSCs

MDSCs suppress immunity mainly through preventing T-cell activation and function, Arg1 and ROS are the common molecules used. Besides, they downregulate macrophage production of the type I cytokine IL-12 to polarize macrophages toward a tumor-promoting phenotype [109,110], suppress tumor cell lysis mediated by NK cells and induce and recruit Tregs [111,112,113,114]. In the presence of MDSCs, the levels of PD-1 expression show a decrease, while PD-L1 expression shows an increase [115]. MDSC-targeted therapy, which decreases MDSC frequency and transforms its function, is studied to overcome the resistance to immune checkpoint inhibitors, thus combining MDSC-targeted therapy and immune checkpoint blockades is considered a promising strategy for the future [116].

### 5.3. TAMs

Protumor macrophages are differentiated through interaction with tumor cells and turn to polarize into M2-like TAM, which play a significant role in immunosuppression, invasion and metastasis. For the sake of overcoming the latent resistance of macrophages, CSF-1R blockade reduces the frequency of TAMs, therefore increasing production of interferon and tumor regression [117], and synergizing with immune checkpoint blockades [118].

### 5.4. IDO

Indole 2,3-dioxygenase is generated by tumors and immune cells to enhance Tregs and MDSCs production and activity. IDO, an enzyme catalyzing the degradation of tryptophan along the kynurenine pathway, is induced in response to inflammatory stimuli and its activity is known to have an inhibition of effector on T-cell immunity [119]. A report conducted on B16 melanoma demonstrated that following PD-1 blockade treatment, a subset of mice with IDO knockout had an obviously slower tumor development and better overall survival rates compared with wild type [120]. Thus, a combination therapy of IDO inhibitors and PD-1/PD-L1 antibodies may demonstrate a better efficacy than single agent [121].

### 5.5. VEGFA

TMB with hypoxia and hyper-angiogenesis is obviously crucial for tumor growth and progression, and vascular endothelial growth factor A (VEGFA) plays a significant role in it. High expression of VEGFA is reported to impair infiltration of effective anti-tumor T-cells, thus leads to innate resistance in PD-1/PD-L1 blockade [122]. Unfortunately, combining treatment with inhibiting the VEGFA and PD-1/PD-L1 blockade demonstrates more toxic and harbors more adverse effects than monotherapy.

### 5.6. Immunosuppressive Cytokines

TGF-β inhibits the expansion and function of many components of the immune system, either by stimulating or inhibiting their differentiation and function, therefore it maintains immune homeostasis and tolerance. Specific chemokines are capable of recruiting cells into tumors. CXCL9, CXCL10, CXCL11, CCL3, CCL4 and other chemokines and their receptors are recruited to cause antitumor response via recruiting CTL and NK cells while CCL2 CCL22, CCL5, CCL7 and CXCL8 recruit immunosuppressive cells to suppress the immune response. Research reveals that epigenetic silencing of CXCL9 and CXCL10 can suppresses T-cell homing [123].

## 6. Activation of Alternative Immune Checkpoints

As one of the most prospective approaches in cancer treatment, immunotherapy has reached notable achievements, especially with the PD-L1 blockade. However, the efficacy of PD-L1 inhibitor therapy has been found to be limited due to activation of other immune checkpoints including TIM-3 and VISTA. So far, some studies have reported that the combination therapy targeting distinct types of immune checkpoints has been proved effective in several cancers.

### 6.1. TIM-3

T-cell immunoglobulin mucin 3 (TIM-3) has been identified as a critical regulator of CTL exhaustion with co-expression of PD-1 [124]. TILs with co-expression of TIM-3 and PD-1 do not produce IL-2 and IFN-γ, and they are prone to exhaust. In response to radiotherapy and PD-L1 inhibition, TIM-3 is upregulated and subsequently caused acquired resistance in HNSCC [107]. Combination therapy targeting TIM-3 and PD-1 signaling pathways simultaneously is proved to be effective against cancer [124].

### 6.2. HHLA2

HHLA2, a member of the B7 family, can predict poor overall survival in several cancers, including human clear cell renal cell carcinoma and colorectal carcinoma [125]. HHLA2 can suppress T-cell activation and proliferation in the presence of TCR and CD28 signaling [126], and can do this more robustly than PD-L1 [127].

### 6.3. VISTA

V-domain Ig suppressor of T-cell activation (VISTA) expression induced by IL-10 and IFN-γ is observed to be higher in immature DCs, MDSCs and Tregs compared with peripheral tissues [128,129]. The synergistic effect of the combining VISTA and PD-L1 monoclonal therapy in colon cancer can be taken as an example, reduction of tumor growth and better OS are observed compared with monotherapy [130].

### 6.4. LAG-3

Lymphocyte activation gene-3 (LAG-3) is responsible for maintaining immune homeostasis through repressing activation of T-cells and cytokines secretion [131]. Interaction between LAG-3 and Galectin-3, a soluble lectin regulating antigen-specific T-cell activation, expands the immunomodulatory effect of LAG-3 on tumor-infiltrating CD8^+^ T-cells in the TME [132]. Sinusoidal endothelial cell lectin binds to LAG-3 to reduce IFN-γ expression produced by activated T-cells [133]. An amazing synergistic effect in suppressing immune responses is found in LAG-3 with PD-1 under distinct conditions [134].

### 6.5. CTLA-4

CD28 interacting with the CD80 dimer and the CD86 monomer mediates T-cell co-stimulation along with TCR signals, while CTLA-4 demonstrates a higher affinity and avidity in conjunction with the two ligands than with CD28, which in turn antagonizes CD28-mediated co-stimulation [135]. A combination of PD-1-targeted mAb nivolumab and CTLA-4-targeted mAb ipilimumab has been approved as the first-line treatment for renal clear cell cancer patients with moderate or poor prognosis [136].

### 6.6. Siglec-15

As a member of the sialic acid-binding immunoglobulin-like lectin (Siglec) gene family, Siglec-15 is found to impair anti-tumor immunity through suppressing T-cell functions. Siglec-15 is expressed only on some myeloid cells normally, while it is upregulated on TAMs and tumor cells [137]. Interestingly, an antagonistic relationship between Siglec-15 and PD-L1 has been reported, mainly due to regulation of IFN-γ [138]. M-CSF induces expression of Siglec-15 on macrophages and IFN-γ, identified as a crucial factor promoting PD-L1 expression, inversely decreases it [137].

### 6.7. TIGHT

T-cell immunoglobulin and ITIM domain (TIGIT), expressed mainly on Tregs, is a co-inhibitory checkpoint receptor which has a significantly higher affinity in binding to CD155 than the co-stimulatory receptor CD226 [139]. TIGIT/CD155 signaling causes T-cell exhaustion to impair anti-tumor immunity in several types of cancer, including melanoma and HNSCC [140,141]. Furthermore, the phenomenon that TIGIT expression often accompanies PD-1 has been observed in both normal tissues and tumors [142].

### 6.8. BTLA

B and T lymphocyte attenuator (BTLA), expressed mostly on B-cells, is upregulated on CD19^+^ high B-cells through AKT and STAT3 pathways once triggered by IL-6 and IL-10 [143]. BTLA is regarded as one of the factors leading to resistance to anti-PD-1 therapy, though they do not suppress T-cell signaling through an identical mechanism related with src-homology-2 domain-containing phosphatase (SHP)1 and SHP2 [143,144].

## 7. Current Combination Therapies with PD-1/PD-L1 Inhibitors

With regard to clinical the limitations of anti-PD-1/PD-L1 monotherapy, it exists more and more in combination therapies based on mechanisms underlying resistance to the PD-1/PD-L1 blockade. Among all of them, chemotherapy, VEGF/VEGFR-targeted therapy and anti-CTLA-4 rank in the top three. Other treatments that are considered to combine with PD-1/PD-L1 blockade include radiotherapy, vaccines, cytokine therapy and chemokine inhibition. Radiotherapy is identified to alter differentiation and function of T-cells and promote the expression of PD-L1, which means adding radiotherapy may enhance the effects of anti-PD-L1 treatment [145]. A triple therapy with anti-PD-1 antagonist antibody, anti-CD137 agonist antibody and vaccine therapy has been reported to significantly enhance T-cell activation in pancreatic ductal adenocarcinoma in a preclinical study [146]. Recently, another immune checkpoint inhibitor tiragolumab targeting TIGIT has been granted breakthrough therapy designation by the FDA and combining anti-PD-L1 and anti-TIGIT has been reported as highly effective in clinic with metastatic NSCLC patients [147]. Combining TNF-α-loaded liposomes and anti-PD-1/PD-L1 further enhances the anti-tumor immunity [148]. Even utilizing newly emerged neoantigens may improve the therapeutic efficacy of immune checkpoint blockade treatment [148].

## 8. Conclusions

As an inhibitor in the immune system, PD-L1 plays multiple roles in tumors. PD-L1 has been confirmed as a prospective and prognostic biomarker in certain cancers, while rebiopsy should be considered when PD-L1 expression is increased due to treatment (such as gefitinib treatment). Immune resistance induced by PD-L1 following various therapies inspired a combination therapy of PD-L1 blockade and these therapies. To date, immunotherapy, especially PD-1/PD-L1 blockade, which is at forefront of clinical therapy, has benefited many patients. However, primary and acquired resistance to this blockade therapy still exists and limits its efficacy. So far, key mechanisms suggest complement approaches for patients who cannot respond well to PD-1/PD-L1 antibodies. For example, modulating the immunosuppressive tumor microenvironment, such as depletion of Tregs, IDO, or MDSCs, interfering suppressive cytokines and inhibiting alternative immune checkpoints, may enhance the therapeutic efficacy of the PD-1/PD-L1 blockade. Other mechanisms underlying resistance to this blockade therapy and individual treatments for more patients requires further investigation.

## Figures and Tables

**Figure 1 cancers-13-00663-f001:**
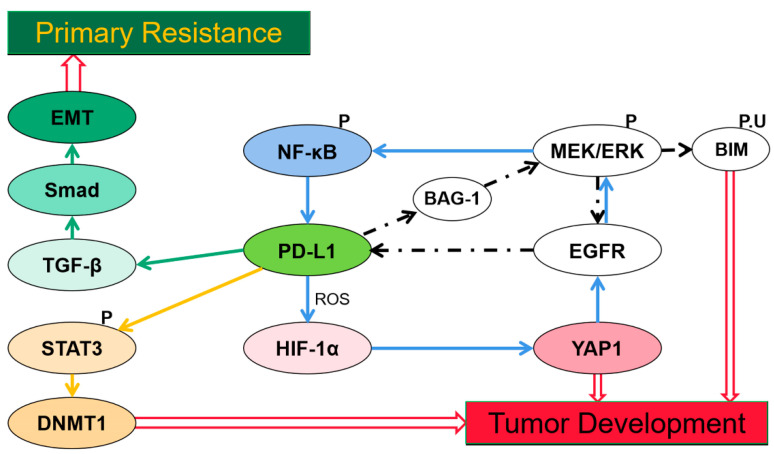
Signaling pathways and factors involved in programmed cell death ligand-1(PD-L1)-induced resistance. (1) PD-L1 expression induced by epidermal growth factor receptor (EGFR) mutation activation via extracellular single-regulated kinase (ERK) signaling, indirectly promotes expression of Bcl-2-associated athanogene-1 (BAG-1), the EGFR/ERK/PD-L1/BAG-1 feedback loop reaches the reactivation of ERK signaling which promotes Bcl-2-interacting mediator of cell death (BIM) phosphorylation to help cells escape from apoptosis. (2) PD-L1-induced hypoxia-inducible factor-1α (HIF-1α) expression is stimulated by reactive oxygen species (ROS), hypoxia increases YAP-1 expression which confers resistance via a YAP1/EGFR/ERK/NF-κB loop. (3) PD-L1 regulates DNA methyltransferase 1(DNMT1) via Signal transducer and activator of transcription 3 (STAT3) signaling and thus induces DNMT1-dependent DNA hypomethylation which promotes cancer development. (4) Activation of transforming growth factor-beta (TGF-β)/Smad pathway induced by PD-L1 is crucial in epithelial-to-mesenchymal transition (EMT) expression which leads to resistance to TKIs.

**Figure 2 cancers-13-00663-f002:**
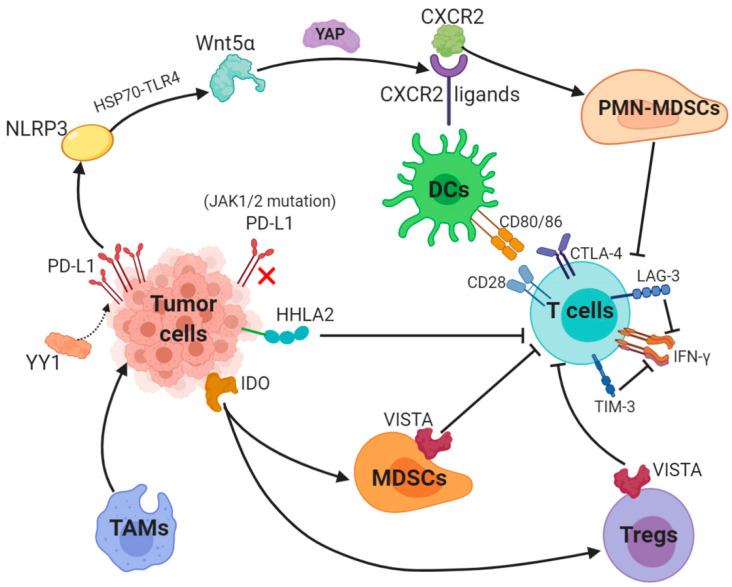
Key mechanisms underlying resistance to PD-L1 (1). The transcription factor Yin Yang 1 (YY1)-induced upregulation of PD-L1 expression triggers NOD-, LRR- and pyrin domain-containing 3 (NLRP3) inflammasome to promote tumor Wnt5α expression via HSP70-TLR4 signaling, and non-canonical WNT ligands activate the YAP pathway to induce chemokine (C-X-C motif) receptor 2 (CXCR2) ligands, while granulocytic subset of myeloid-derived suppressor cells (PMN-MDSCs) relied on CXCR2 to suppress T-cell function. (2) Loss-of-function mutations in JAK1/2 leads to the paucity of PD-L1 expression. (3) Tumor-suppressing microenvironment. Tumor-associated macrophages (TAMs) promote tumor progression, while Indole 2,3-dioxygenase (IDO) generated by tumors enhances Tregs and MDSCs activity, which suppress immunity. (4) Activation of alternative immune checkpoints. T-cell immunoglobulin mucin 3 (TIM-3) and Lymphocyte activation gene-3 (LAG-3) produced by T-cells impair generation of IFN-γ, which activates T-cells. CTLA-4 demonstrates a higher affinity and avidity in conjunction with CD80 and CD86 than CD28 to antagonize costimulation. VISTA is found to be related to MDSC mainly derived CD33 expression and HHLA2 decreases T-cell proliferation.

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
