# Peer review of "Recent Advancements in the Mechanisms Underlying Resistance to PD-1/PD-L1 Blockade Immunotherapy"

_cancers, 2021, doi:10.3390/cancers13040663_

Round 1
Reviewer 1 Report
The manuscript may be accepted in the present form
Reviewer 2 Report
I stand with my previous review. With reference to my previous review the new manuscript changes are not substantial. The review remains same except addition of few lines and references. Though the analysis is good but novel concept is missing, as will as authors dont have substantial work or published articles in this field.
Reviewer 3 Report
All the concerns have been addressed.
This manuscript is a resubmission of an earlier submission. The following is a list of the peer review reports and author responses from that submission.
Round 1
Reviewer 1 Report
Remarks to the authors:
In this manuscript, the authors have demonstrated the mechanism of resistance in PD-1-blockade or/and PD-L1-blockade immunotherapy. In addition, the authors summarized combination of therapeutic strategies with PD-1/PD-L1 blockade to overcome the limitation of this therapy. This review is well structured as well as described, but the authors need to add additional contents about combination therapy with PD-1/PD-L1 blockade. In addition, there are several points that need to be revised.
Major Comments
- In page 2, the authors mentioned as “So far, five immune checkpoint inhibitors targeting PD-1/PD-L1 have been…targeting PD-L1”. In September 2018, Cemiplimab targeting PD-1 was approved by the US Food and Drug Administration. Therefore, it is recommended to add this monoclonal antibody against PD-1.
- In Figure 2, CXCR2 was placed between Wnt5a and PMN-MDSCs. However, the authors mentioned as “…non-canonical WNT ligands activate YAP pathway to induce CXCR2 ligands, while PMN-MDSCs relied on CXCR2 suppress T-cell function” in the figure legend. Therefore, it would be better to change CXCR2 to CXCR2 ligands and PMN-MDSCs to CXCR2-expressing PMN-MDSCs.
- In “5.1 Tregs”, the authors mentioned as “Combination of radiation therapy and dual immune checkpoint blockade restores antitumor immunity of consumed Tregs”. Previous studies have demonstrated that combining anti-CD25 therapy with PD-1 blockade induces significant therapeutic effect through Treg depletion. Here, the authors can add anti-CD25 antibody as a combination of therapeutic strategy with immune checkpoint blockade.
- Wanling Chen et al., 2020 had reported that MDSC-targeted therapy is a promising strategy to overcome the resistance to immune checkpoint inhibitors. In that review, they have discussed the MDSC targeting methods as reduction of MDSC frequency or alteration of MDSC function. Therefore, it would be better to add the combination therapy of MDSC-targeted method with immune checkpoint blockade in “5.2 MDSCs”.
- It would be better to add section before conclusion section; 'combination therapies to overcome the resistance in PD-1 or/and PD-L1 therapy' and 'result from clinical trials' for those approaches.
Minor Comments
- Please add abbreviation in the figure legend.
- The authors need to change “+” of “CD8+” and “CD103+” to superscript form.
Author Response
Major comments:
- In page 2, the authors mentioned as “So far, five immune checkpoint inhibitors targeting PD-1/PD-L1 have been…targeting PD-L1”. In September 2018, Cemiplimab targeting PD-1 was approved by the US Food and Drug Administration. Therefore, it is recommended to add this monoclonal antibody against PD-1.
Reply: Thank your advices, and we have made complements with “Cemiplimab”.
- In Figure 2, CXCR2 was placed between Wnt5a and PMN-MDSCs. However, the authors mentioned as “…non-canonical WNT ligands activate YAP pathway to induce CXCR2 ligands, while PMN-MDSCs relied on CXCR2 suppress T-cell function” in the figure legend. Therefore, it would be better to change CXCR2 to CXCR2 ligands and PMN-MDSCs to CXCR2-expressing PMN-MDSCs.
Reply: Thank your advices, and we have revised these expressions in the Figure 2.
- In “5.1 Tregs”, the authors mentioned as “Combination of radiation therapy and dual immune checkpoint blockade restores antitumor immunity of consumed Tregs”. Previous studies have demonstrated that combining anti-CD25 therapy with PD-1 blockade induces significant therapeutic effect through Treg depletion. Here, the authors can add anti-CD25 antibody as a combination of therapeutic strategy with immune checkpoint blockade.
Reply: Thank your advices, and we have made complements with “anti-CD25 antibody”
- Wanling Chen et al., 2020 had reported that MDSC-targeted therapy is a promising strategy to overcome the resistance to immune checkpoint inhibitors. In that review, they have discussed the MDSC targeting methods as reduction of MDSC frequency or alteration of MDSC function. Therefore, it would be better to add the combination therapy of MDSC-targeted method with immune checkpoint blockade in “5.2 MDSCs”.
Reply: Thank your advices, and we have made complements with “the combination therapy of MDSC-targeted method”
- It would be better to add section before conclusion section; 'combination therapies to overcome the resistance in PD-1 or/and PD-L1 therapy' and 'result from clinical trials' for those approaches.
Reply: Thank your advices, and we added a section named ” Current combination therapies with PD-1/PD-L1 inhibitors” before conclusion section, we discussed different combination of therapies that are utilized for combining with PD-1/PD-L1 blockade therapy including chemotherapy, anti-CTLA-4, vaccine therapy. Besides, we have studied some latest findings including combination of anti-TIGIT and anti-PD-L1 therapy.
Minor Comments
- Please add abbreviation in the figure legend.
Reply: Thank your advices, and we have revised it.
- The authors need to change “+” of “CD8+” and “CD103+” to superscript form.
Reply: Thank your advices, and we have revised it.
Reviewer 2 Report
The manuscript entitled “Recent advancements in the mechanisms underlying resistance to PD-1/PD-L1 blockade immunotherapy” by Yuan et al describes mechanisms underlying resistance to PD-1/PD-L1 blockade immunotherapy recently. In this review, authors have discussed PD-1/PD-L1 expression associated immune escape and tumor progression. However similar reviews were published recently like Sun et al. Biomarker Research 2020 [PMID: 32864132], where authors have mechanisms underlying the resistance to PD-1/PD-L1 blockade: The lack of tumor antigens, T cell dysfunction, PD-1 or PD-L1, Noncoding RNAs and Gut microbiome; Ren et al Mol Cancer 2020 [PMID: 32000802] discussed predictive biomarkers of the efficacy of PD-1 blockade therapy; while other reviews focused on specific cancer type such as in colorectal cancer [Yaghoubi, 2019; PMID:30522017], hepatocellular carcinoma [Li 2020l PMID: 32547550] and so on. Authors did great job to discuss the literature however with current published reviews there still missing novel information for readers.
Thus to reject in Cancers but can consider for other sister journals.
Author Response
General comments
The manuscript entitled “Recent advancements in the mechanisms underlying resistance to PD-1/PD-L1 blockade immunotherapy” by Yuan et al describes mechanisms underlying resistance to PD-1/PD-L1 blockade immunotherapy recently. In this review, authors have discussed PD-1/PD-L1 expression associated immune escape and tumor progression. However similar reviews were published recently like Sun et al. Biomarker Research 2020 [PMID: 32864132], where authors have mechanisms underlying the resistance to PD-1/PD-L1 blockade: The lack of tumor antigens, T cell dysfunction, PD-1 or PD-L1, Noncoding RNAs and Gut microbiome; Ren et al Mol Cancer 2020 [PMID: 32000802] discussed predictive biomarkers of the efficacy of PD-1 blockade therapy; while other reviews focused on specific cancer type such as in colorectal cancer [Yaghoubi, 2019; PMID:30522017], hepatocellular carcinoma [Li 2020l PMID: 32547550] and so on. Authors did great job to discuss the literature however with current published reviews there still missing novel information for readers.
Reply: Thank your advices, similar reviews about the resistance to PD-1/PD-L1 blockade had been published recently, which means research hotpot and is valuable to update any new findings for the readers. And we have found more recent studies for our review. In the paragraph 4.2, we added “the regulating effect of NAD+ metabolism on PD-L1 expression.” In the paragraph 4.3, we added “HLA loss may contribute to acquire resistance during immunotherapy.” In the paragraph 5.5, we made a new section for elucidating the influence of VEGFA in leading to innate resistance to PD-1/PD-L1 blockade.” And in paragraph 7, we discussed several latest findings in combination therapies with PD-1/PD-L1 inhibitors.”
Reviewer 3 Report
The manuscript entitled:" Recent advancements in the mechanisms underlying resistance to PD-1/PD-L1 blockade immunotherapy" focused on a systemic revision of literature data about the identification of resistance mechanism that allows a modulation to clinical response for PD-L1 inibhitors is well written and requires some minor revisions to be suitable for publication:
- In the introduction section, the authors should also report the clinical stage where IC may be adopted. In my opinion, this data may improve th quality of the manuscript by elucidating the clinical setting for their adoption.
- In the paragraph 2, the authors well described some minor examples given to explain clinical outcome in NSCLC patients harboring both EGFR sensitive mutations and PD-L1 expression. In my opinion, this critical issue should be better elucidated by the authors.
- In the paragraph 3.2, the authors describe the most relevant molecular mechanisms behind the development of acquired resistance to IC. According to this point, the authors should elucidate if new drugs or therapeutic strategies were under investigation to overcome this limitation. In addition, the authors should better define the setting where these biomarkers were evaluated ( pre clinical or clinical studies) in order to clarify the state of art about these novel data.
Author Response
General comments
The manuscript entitled:" Recent advancements in the mechanisms underlying resistance to PD-1/PD-L1 blockade immunotherapy" focused on a systemic revision of literature data about the identification of resistance mechanism that allows a modulation to clinical response for PD-L1 inibhitors is well written and requires some minor revisions to be suitable for publication.
- In the introduction section, the authors should also report the clinical stage where IC may be adopted. In my opinion, this data may improve the quality of the manuscript by elucidating the clinical setting for their adoption.
Reply: Thank your advices, and we have added more detailed applications for the immunotherapy, including “Immunotherapy has received the US FDA approval for 57 indications in 17 solid tumors in less than 10 years, while over 80% are PD-1/PD-L1-targeted antibodies.”
- In the paragraph 2, the authors well described some minor examples given to explain clinical outcome in NSCLC patients harboring both EGFR sensitive mutations and PD-L1 expression. In my opinion, this critical issue should be better elucidated by the authors.
Reply: Thank your advices, and we have added more detailed information about EGFR sensitive mutations and PD-L1 expressions, besides, we searched a recent study for clinical outcome of combining anti-PD-L1 and gefitinib”
- In the paragraph 3.2, the authors describe the most relevant molecular mechanisms behind the development of acquired resistance to IC. According to this point, the authors should elucidate if new drugs or therapeutic strategies were under investigation to overcome this limitation. In addition, the authors should better define the setting where these biomarkers were evaluated ( preclinical or clinical studies) in order to clarify the state of art about these novel data.
Reply: Thank your advices, and we have found the findings of whether new drugs and combination therapies have been under investigation, and we have added these information below each paragraphs.